# Cytomegalovirus reactivation under pre-emptive therapy following allogeneic hematopoietic stem cell transplant: Pattern, survival, and risk factors in the Republic of Korea

**Ka-Won Kang, Min Ji Jeon, Eun Sang Yu, Dae Sik Kim, Byung-Hyun Lee, Se Ryeon Lee, Chul Won Choi, Yong Park, Byung Soo Kim, Hwa Jung Sung** *

Division of Hematology-Oncology, Department of Internal Medicine, Korea University College of Medicine, Seoul, Republic of Korea

* doctorsung@korea.ac.kr

## Abstract

### Introduction

Pre-emptive therapy for cytomegalovirus (CMV) reactivation has been used in allogeneic hematopoietic stem cell transplantation (allo-HSCT). It is unclear if this strategy has poorer clinical outcomes in CMV-endemic areas and if more aggressive prophylaxis is required.

### Methods

We retrospectively analyzed the patterns and survival after CMV reactivation in patients undergoing pre-emptive therapy following allo-HSCT and assessed high-risk patients who could benefit from aggressive CMV prophylaxis in endemic areas.

### Results

Of the 292 patients who underwent allo-HSCT, 70.5% (donor+ or recipient+) were CMV seropositive. CMV reactivation occurred in 139 patients (47.6%), with a median of 31.5 days from day 0 of allo-HSCT. The overall survival of patients with CMV reactivation who received pre-emptive therapy did not differ from those without reactivation. Of the 139 patients with CMV reactivation, 78 (56.1%) underwent ≥2 rounds of pre-emptive therapy. In multivariate analysis, the risk of CMV reactivation was higher in patients with multiple myeloma, with CMV seropositivity of the recipient and donor, administered with a higher dose of anti-thymocyte globulin (ATG), and with acute graft-versus-host disease (aGVHD) ≥ grade 2.

### Conclusion

Although half of the patients with allo-HSCT were administered with pre-emptive therapy for CMV, CMV reactivation did not affect their survival, indicating the advantages of pre-

**Data Availability Statement:** All relevant data are within the paper and its Supporting Information files.

**Funding:** This research was supported by a Korea University Grant (number: K2125711). The funders had no role in study design, data collection and analysis, decision to publish, or preparation of the manuscript.

**Competing interests:** The authors have declared that no competing interests exist.

emptive therapy, even in CMV-endemic areas. The cost-effectiveness of more aggressive CMV prophylaxis should be re-evaluated in patients at a high risk for CMV reactivation.

## Introduction

Cytomegalovirus (CMV) is endemic to the Republic of Korea. Endemic areas are generally identified according to CMV seroprevalence, defined as the prevalence of anti-CMV immuno-globulin G (IgG) antibodies in the serum within a given population. The estimated CMV sero-prevalence in Koreans has been approximately 90% over the past two decades [1–3], which is a relatively high number compared to that in other countries, including Germany (about 40%), the United States (about 50%), and Japan (about 65%) [4–8]. Although not all CMV-seropositive recipients of allogeneic hematopoietic stem cell transplantation (allo-HSCT) experience CMV reactivation, the risk is higher in CMV-seropositive recipients placed with grafts from CMV-seronegative donors due to lack of donor-transferred CMV-specific immunity during host reactivation of endogenous latent CMV. Moreover, CMV reactivation is associated with high morbidity and mortality in recipients of allo-HSCT [9–12].

CMV primary prophylaxis is recommended in recipients of allo-HSCT at high risk of CMV reactivation following allo-HSCT, with letermovir being considered in CMV-seropositive recipients. In a phase 3 trial, letermovir prophylaxis gave rise to a significantly lower risk of CMV infection compared to placebo in patients at high risk for CMV-related disease undergoing allo-HSCT and who presented with CMV seropositivity [13]. The efficacy of letermovir prophylaxis in preventing CMV reactivation is generally well-accepted in the real-world [14–16]. Based on these data, letermovir should be considered a primary prophylaxis in most allo-HSCT recipients in the Republic of Korea. However, there are insufficient data on whether the conventional pre-emptive therapy for CMV has poor clinical outcomes in CMV-endemic areas and whether this pre-emptive strategy should be changed to more aggressive prophylaxis in all patients. Therefore, objective information is required to identify the advantages and dis-advantages of the existing CMV preemptive therapy for confirming that letermovir should be used in most asymptomatic high-risk patients in CMV endemic areas.

In this study, we retrospectively analyzed the patterns and survival of CMV reactivation in patients undergoing pre-emptive therapy without using letermovir for CMV primary prophy-laxis following HSCT and high-risk patients who could benefit from aggressive primary or sec-ondary CMV prophylaxis in CMV-endemic area should be assessed.

## Materials and methods

### Patients

We retrospectively analyzed data from three tertiary hospitals affiliated with the Korea University Medical Center (Anam, Guro, and Ansan Hospitals) in the Korea University Transplant Registry from November 2003 to July 2020. Patients who met all the following criteria were included: (1) had undergone allo-HSCT; (2) were monitored for CMV reactivation using CMV antigen or CMV polymerase chain reaction (PCR) for at least 6 months from the date of allo-HSCT; (3) were treated with pre-emptive therapy for CMV reactivation. Patients who used letermovir for CMV primary prophylaxis, whose first CMV reactivation occurred 6 months after allo-HSCT, and who received salvage treatment for disease recurrence before the first CMV reactivation were excluded. All observable periods were checked to include

recurrent CMV reactivation after the first CMV reactivation. Data were originally collected between December 2020 and December 2021.

All procedures involving human participants were performed in accordance with the ethical standards of the institutional and national research committees and the 1964 Helsinki Declaration and its later amendments or comparable ethical standards. The sex, age, and medical information of each patient were obtained, but personal information was not collected. All information was anonymized to ensure that individual participants could not be identified. The study was approved by the Institutional Review Board (IRB) of the Korea University Medical Center, and all data were fully anonymized (Anam Hospital: IRB No. 2020AN0444, Guro Hospital: IRB No. 2020GR0505, and Ansan Hospital: IRB No. 2020AS0343). As this study was conducted using anonymous patient data, the requirement for informed consent was waived by the IRB.

## Transplantation

The non-myeloablative conditioning regimen consisted of intravenous cyclophosphamide (Cy) (50 mg/kg of body weight, from day -5 to day -2) or fludarabine (Flu) (30 mg/m$^2$ of body surface area (BSA), from day -7 to day -2), or Cy (300 mg/m$^2$ of BSA, from day -6 to day -3) and Flu (30 mg/m$^2$ of BSA, from day -6 to day -3). The reduced-intensity conditioning regimen consisted of intravenous busulfan (Bu) (3.2 mg/kg of body weight, days -7 and -6) and Flu (30 mg/m$^2$ of BSA, from day -7 to day -2). The myeloablative conditioning regimen consisted of intravenous Bu (3.2 mg/kg of body weight, from day -7 to day -4) and Flu (30 mg/m$^2$ of BSA, from day -7 to day -2), or Bu (3.2 mg/kg of body weight, from day -7 to day -4) and Cy (60 mg/kg of body weight, days -3 and -2), or total-body irradiation (3 Gy from days -7 to -4) and Cy (60 mg/kg of body weight, days -3 and -2). Patients who underwent T cell depletion were excluded from the study.

All patients received cyclosporine (3 mg/kg of body weight) or tacrolimus (0.03 mg/kg of body weight) combined with methotrexate (15 mg/m$^2$ of BSA on day 1 and 10 mg/m$^2$ on days 3, 6, and 11) for graft-versus-host disease (GVHD) prophylaxis, and levofloxacin (500 mg, once daily), acyclovir (400 mg, twice daily), micafungin (50 mg, once daily), and sulfamethoxazole/trimethoprim (400/80 mg, twice daily) for infection prophylaxis. The use and dose of rabbit anti-thymocyte globulin (ATG) for the prevention of GVHD during conditioning were determined at the discretion of the participating physicians according to the type of donor or conditioning regimen.

## CMV monitoring and pre-emptive therapy for CMV reactivation during allo-HSCT

CMV monitoring was performed for all enrolled patients using CMV antigen (once or twice weekly from 2003 to 2013) or CMV PCR (once or twice weekly from 2013 to 2020) for at least 6 months from the date of allo-HSCT. The presence of the CMV pp65 antigen was analyzed using Clonab CMV® pp65 (Bio-Rad, Germany), and CMV PCR was performed using an Artus ® CMV QS-RGQ kit (Qiagen, Germany).

Patients with positive CMV antigenemia or a CMV PCR titer >1,000 copies/mL received pre-emptive therapy for CMV by administration of continuous intravenous injection of ganciclovir (5 mg/kg of actual body weight every 12 h) until a negative test result for the CMV antigen or a CMV PCR titer of <1,000 copies/mL was noted according to the treatment policy of the Korea University Medical Center.

## Clinical endpoints

The primary endpoints were the patterns and survival of patients with CMV reactivation who had undergone CMV monitoring and pre-emptive therapy for CMV infection during allo-HSCT in a CMV-endemic area. The secondary endpoint was the investigation of high-risk patients who could benefit from aggressive primary or secondary CMV prophylaxis. The following data were collected: age, sex, diagnosis, type of donor (matched, mismatched at one allele, or haploidentical), hematopoietic cell transplantation-specific comorbidity index (HCT-CI score) [17], CMV serostatus of the donor and recipient, type of conditioning regimen (non-myeloablative, reduced intensity, or myeloablative), ATG use and dose, type of immunosuppressant (cyclosporine or tacrolimus), the occurrence of acute GVHD grade ≥2 or chronic GVHD with more than moderate severity, and recurrence of the disease. The occurrence of acute GVHD grade ≥2 or chronic GVHD with more than moderate severity was distinguished using an operational definition. According to the general guidelines of GVHD and treatment policy of the Korea University Medical Center, we had been treating patients with acute GVHD grade ≥2 or chronic GVHD with more than moderate severity using systemic steroid therapy with a dosage of at least 1 mg/kg of actual bodyweight or higher for at least 7 days [18–22]. Therefore, we classified patients with acute GVHD grade ≥2 or chronic GVHD with more than moderate severity as those who received systemic steroid therapy with an aforementioned dosage. Acute GVHD was classified as patients who met the aforementioned criteria within 100 days from the transplantation date, while chronic GVHD was classified as patients who met the aforementioned criteria after 100 days.

## Statistical analysis

Median values and ranges were reported for continuous variables, and percentages were reported for categorical values. Categorical values were analyzed using the chi-squared test. Overall survival (OS) was defined as the time from day 0 of allo-HSCT to death from any cause or censoring. Factors affecting OS or CMV reactivation were assessed using the Cox proportional hazards model for univariate and multivariate analyses. OS was analyzed using the following variables: age, sex, diagnosis, type of donor, HCT-CI score, type of conditioning regimen, ATG use, type of immunosuppressant, CMV reactivation after allo-HSCT, the occurrence of acute GVHD grade ≥2 or chronic GVHD with more than moderate severity during all observable periods, and recurrence of the disease. The risk of the first CMV reactivation was analyzed using the following variables: age, sex, diagnosis, type of donor, HCT-CI score, CMV serostatus of the donor and recipient, type of conditioning regimen, ATG use and dose, type of immunosuppressant, the occurrence of acute GVHD grade ≥2, and the occurrence of chronic GVHD with more than moderate severity within six months after the date of allo-HSCT. IBM SPSS Statistics for Windows, version 21.0 (IBM Corp., Armonk, NY, USA) was used for data analysis. *P* <0.05 was considered statistically significant.

## Results

### Patient characteristics

A total of 292 patients who underwent allo-HSCT were analyzed (**Table 1**). The median age was 46.5 years (range: 16.0–68.0 years), and the diagnoses were aplastic anemia (26 patients, 8.9%), acute myeloid leukemia/myelodysplastic syndrome (180 patients, 61.6%), acute lymphoblastic leukemia (64 patients, 21.9%), multiple myeloma (11 patients, 3.8%), and lymphoma (11 patients, 3.8%). CMV serostatus was confirmed in all patients in this study, with CMV seropositivity of the recipient or donor at 70.5% (donor-recipient+: 21 patients, 7.2%;

**Table 1. Baseline patient characteristics at the time of allogeneic hematopoietic stem cell transplantation.**

| Characteristic | Total patients (n = 292) |
|---|---|
| Median age, years (range) | 46.5 (16.0–68.0) |
| <50 years, n (%) | 177 (60.6) |
| ≥50 years, n (%) | 115 (39.4) |
| Sex | |
| Male, n (%) | 163 (55.8) |
| Female, n (%) | 1294 (44.2) |
| Diagnosis | |
| Aplastic anemia | 26 (8.9) |
| Acute myeloid leukemia/myelodysplastic syndrome | 180 (61.6) |
| Acute lymphoblastic leukemia | 64 (21.9) |
| Multiple myeloma | 11 (3.8) |
| Lymphoma | 11 (3.8) |
| Donor type | |
| Matched | 222 (76.0) |
| Mismatched at one allele | 13 (4.5) |
| Haploidentical | 57 (19.5) |
| HCT-CI score, n (%) | |
| 0 | 111 (38.0) |
| 1–2 | 128 (43.8) |
| ≥3 | 53 (18.2) |
| CMV serostatus, n (%) | |
| Donor-Recipient- | 86 (29.5) |
| Donor-Recipient+ | 21 (7.2) |
| Donor+Recipient+ | 60 (20.5) |
| Donor+Recipient- | 125 (42.8) |
| Type of conditioning regimen, n (%) | |
| Non-myeloablative | 22 (7.5) |
| Reduced intensity | 168 (57.5) |
| Myeloablative | 102 (34.9) |
| Anti-thymocyte globulin use, n (%) | 200 (68.5) |
| <5 mg/kg | 84 (28.8) |
| ≥5 mg/kg and <9 mg/kg | 77 (26.4) |
| ≥9 mg/kg | 39 (13.4) |
| Immunosuppressant use, n (%) | |
| Cyclosporin | 202 (69.2) |
| Tacrolimus | 90 (30.8) |
| Acute GVHD of grade ≥2, n (%) | 99 (33.9) |
| Chronic GVHD of grade ≥ moderate severity, n (%)* | 60 (20.5) |

Note: *Among them, 18 patients experienced chronic GVHD with more than moderate severity within 6 months from the transplantation date.

HCT-CI, hematopoietic cell transplantation-specific comorbidity index; CMV, cytomegalovirus

donor+recipient+: 60 patients, 20.5%; and donor+recipient-: 125 patients, 42.8%). ATG for the prevention of GVHD during conditioning was administered to 68.5% of patients (200/292 patients). ATG doses were classified as <5, ≥5, <9, and ≥9 mg/kg according to the ATG dose frequency of the enrolled patients in this study. The number of patients in each group was

28.8% (84/292 patients), 26.4% (77/292 patients), and 13.4% (39/292 patients). Among 292 patients, acute GVHD grade ≥2 was observed in 99 (33.9%) patients, while chronic GVHD with more than moderate severity was identified in 60 (20.5%) patients.

### Pattern and survival of CMV reactivation in patients undergoing CMV monitoring and pre-emptive therapy during allo-HSCT

CMV reactivation occurred in 139 of 292 patients (47.6%), with a median of 31.5 days (range: 29.0–180.0 days) from the date of allo-HSCT to the first day of pre-emptive therapy for CMV (Fig 1). There was no significant difference in the OS of patients with CMV reactivation compared to that of patients without CMV reactivation (hazard ratio [HR]:0.978, 95% confidence interval [CI]:0.694–1.378, $p = 0.899$) when the following factors were corrected: age, sex, diagnosis, type of donor, HCT-CI score, type of conditioning regimen, ATG use, type of immunosuppressant, CMV reactivation after allo-HSCT, occurrence of acute GVHD grade ≥2 or chronic GVHD with more than moderate severity during all observable periods, and recurrence of the disease (Table 2). OS was independently associated with the occurrence of acute GVHD of grade ≥2 (HR:1.875; 95% CI:1.310–2.684, $p = 0.001$) and disease recurrence (HR:3.223, 95% CI:2.265–4.587, $p <0.001$). Of the patients with CMV reactivation, 43.9% (61/139) died, and the causes of death were all forms of infection (28/61), disease progression (18/

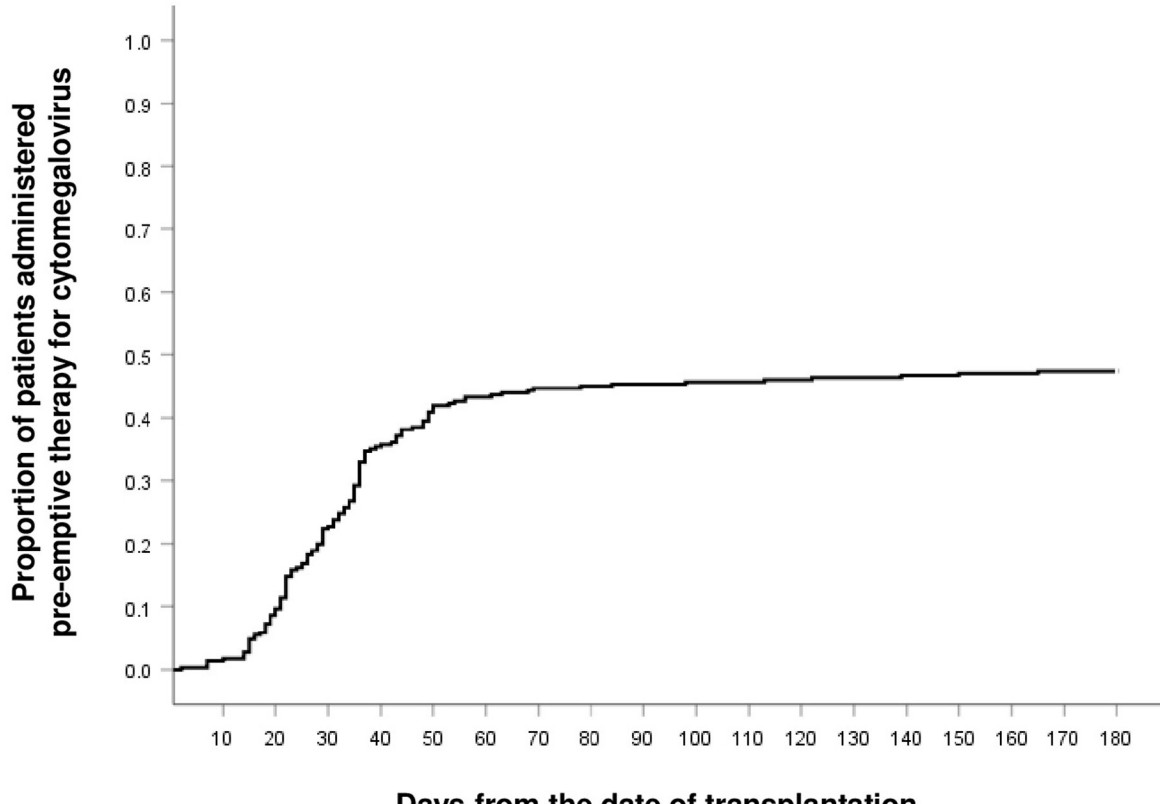

**Fig 1. Time to the start of pre-emptive therapy for CMV reactivation during allo-HSCT in patients positive for CMV antigen or with CMV PCR titers >1,000 copies/mL.** Abbreviations: CMV, cytomegalovirus; allo-HSCT, allogeneic hematopoietic stem cell transplantation; PCR, polymerase chain reaction.

**Table 2. Univariate and multivariate analyses of variables associated with overall survival.**

| Variables | Univariate analysis | | Multivariate analysis | |
|---|---|---|---|---|
| | HR (95% CI) | p-value | OR (95% CI) | p-value |
| Acute GVHD of grade ≥2 versus the others | 1.549 (1.089–2.202) | **0.015** | 1.875 (1.310–2.687) | **0.001** |
| Disease recurrence | 2.919 (2.063–4.131) | <0.001 | 3.223 (2.265–4.587) | <**0.001** |

Note: Bold text indicates statistical significance.

CMV, cytomegalovirus; HCT-CI, hematopoietic cell transplantation-specific comorbidity index; HR, hazard ratio; CI, confidence interval; GVHD, graft-versus-host disease

61), GVHD (12/61), and unknown (3/61). Three patients received pre-emptive therapy for CMV reactivation within 1 week from the date of death. Of the patients without CMV reactivation, 46.4% (71/153) died, and the causes of death were all forms of infection (38/71), disease progression (23/71), GVHD (7/71), and unknown (3/71).

To monitor recurrent CMV reactivation after the first CMV reactivation, all observable periods were checked. The median duration of the first pre-emptive therapy for CMV was 16 days (range: 1–73 days). Of the 139 patients with CMV reactivation, 78 (56.1%) underwent two or more than two rounds of pre-emptive therapy for CMV. The median durations of the second, third, fourth, and fifth or more rounds of pre-emptive therapy for CMV reactivation were ten days (range: 1–44 days), 13 days (range: 1–74 days), ten days (range: 1–54 days), and 15 days (range: 1–44 days), respectively.

## Patients at high risk for CMV reactivation

The risk of the first CMV reactivation was analyzed using the following variables: age, sex, diagnosis, type of donor, HCT-CI score, CMV serostatus of the donor and recipient, type of conditioning regimen, ATG use and dose, type of immunosuppressant, occurrence of acute GVHD grade ≥2, and the occurrence of chronic GVHD with more than moderate severity within six months after the date of allo-HSCT. In the multivariate analysis, the risk of the first CMV reactivation was high in patients with multiple myeloma (HR:3.912, 95% CI:1.619–9.450, $p = 0.002$), recipient and donor CMV seropositivity before allo-HSCT (HR:2.581, 95% CI:1.590–4.191, $p < 0.001$), acute GVHD grade ≥2 (HR:1.680, 95% CI:1.170–2.414, $p = 0.005$), and use of ATG (**Table 3**). In the case of patients taking ATG, the risk increased with an increase in the total dose of ATG (<5 mg/kg, HR:1.972, 95% CI:1.180–3.294, $p = 0.010$; ≥5 mg/kg and <9 mg/kg, HR:2.915, 95% CI:1.800–4.720, $p < 0.001$; and ≥9 mg/kg, HR:6.460, 95% CI:3.766–11.081, $p < 0.001$) compared to that in patients who did not receive ATG. **Fig 2** presents the pattern of the first CMV reactivation according to ATG use and dose after correcting for the abovementioned factors. Approximately half of the patients administered ATG were treated with pre-emptive therapy for CMV (55.0%, 110/200 patients), and the proportion of patients receiving pre-emptive therapy for CMV reactivation increased significantly with the total dose of ATG (ATG dose <5 mg/kg, 39/86 [45.3%] patients, reference; ≥5 and <9 mg/kg, 44/77 [57.1%] patients, p = 0.089; and ≥9 mg/kg, 30/39 [76.9%] patients, p = 0.001).

## Discussion

In this study, 47.6% of the patients showed CMV reactivation and were treated with pre-emptive therapy during allo-HSCT. CMV reactivation and treatment with pre-emptive therapy did not affect the survival of these patients. However, approximately half of these patients experienced CMV reactivation more than twice, with a median treatment duration of 10–15 days.

**Table 3. Univariate and multivariate analyses of variables associated with the first CMV reactivation.**

| Variables | Univariate analysis | | Multivariate analysis | |
|---|---|---|---|---|
| | HR (95% CI) | p-value | HR (95% CI) | p-value |
| Diagnosis | | | | |
| Aplastic anemia | 1 | | 1 | |
| Acute myeloid leukemia/myelodysplastic syndrome | 1.005 (0.560–1.805) | 0.986 | 1.367 (0.755–2.477) | 0.302 |
| Acute lymphoblastic leukemia | 1.129 (0.591–2.158) | 0.714 | 1.512 (0.784–2.918) | 0.217 |
| Multiple myeloma | 2.523 (1.078–5.909) | **0.033** | 3.912 (1.619–9.450) | **0.002** |
| Lymphoma | 0.731 (0.238–2.241) | 0.583 | 1.090 (0.349–3.398) | 0.882 |
| CMV status | | | | |
| Donor-Recipient- | 1 | | 1 | |
| Donor-Recipient+ | 1.215 (0.584–2.528) | 0.602 | 1.235 (0.588–2.594) | 0.577 |
| Donor+Recipient+ | 1.910 (1.199–3.043) | 0.006 | 2.581 (1.590–4.191) | **<0.001** |
| Donor+Recipient- | 1.249 (0.822–1.897) | 0.298 | 1.470 (0.961–2.249) | 0.076 |
| Anti-thymocyte globulin use | | | | |
| No use | 1 | | 1 | |
| <5 mg/kg | 1.630 (0.999–2.658) | 0.050 | 1.972 (1.180–3.294) | **0.010** |
| ≥5 mg/kg and <9 mg/kg | 2.351 (1.470–3.761) | **<0.001** | 2.915 (1.800–4.720) | **<0.001** |
| ≥9 mg/kg | 4.487 (2.684–7.501) | **<0.001** | 6.460 (3.766–11.081) | **<0.001** |
| Acute GVHD of grade ≥2 versus the others | 1.386 (0.987–1.948) | 0.060 | 1.680 (1.170–2.414) | **0.005** |

Note: Bold text indicates statistical significance. *This event was counted only when it occurred within 6 months from the date of the allogeneic hematopoietic stem cell transplant.

CMV, cytomegalovirus; HCT-CI, hematopoietic cell transplantation-specific comorbidity index; HR, hazard ratio; CI, confidence interval; GVHD, graft-versus-host disease

The risk of CMV reactivation was higher in patients with multiple myeloma, CMV seropositivity of recipients and donors before allo-HSCT, acute GVHD grade ≥2, and the use and dose of ATG.

Although the incidence of CMV reactivation can differ according to the situation at the time of transplantation, region, and monitoring methods for CMV, reactivation has been reported to develop in approximately 50% of cases (range: 10–70%) 4–8 weeks after allo-HSCT [23–29]. In this study, 47.6% of patients showed CMV reactivation, most of which occurred within 100 days (median: 31.5 days, range: 29.0–180.0 days). CMV is endemic in the Republic of Korea; however, the incidence of CMV reactivation was comparable to previously reported values. Additionally, CMV reactivation did not affect patient survival in this study, contrary to previous reports that CMV is associated with increased non-recurrence mortality [24, 25]. In the previous studies, the major CMV monitoring method was the detection of the CMV antigen; however, in this study, more than half the patients were monitored using CMV PCR (CMV antigen: 104/292 patients, 35.6%; CMV PCR: 188/292 patients, 64.4%). CMV reactivation can be detected earlier by CMV PCR than by monitoring the CMV antigen [30–32]. Studies performed with PCR-based CMV monitoring during allo-HSCT showed no effect of CMV reactivation on survival [27, 30, 33]. These findings suggest that under proper prophylaxis, CMV monitoring, and pre-emptive therapy for CMV reactivation, CMV reactivation may not affect the survival of patients undergoing allo-HSCT, even in CMV-endemic areas.

However, approximately half of the patients in our study with CMV reactivation experienced ≥ 2 CMV reactivation events, with a median treatment duration of 10–15 days. Previous studies have also reported repeated CMV reactivation in approximately 50% of cases (range: 30–70%) [25, 34, 35]. Although CMV reactivation may not affect the survival of

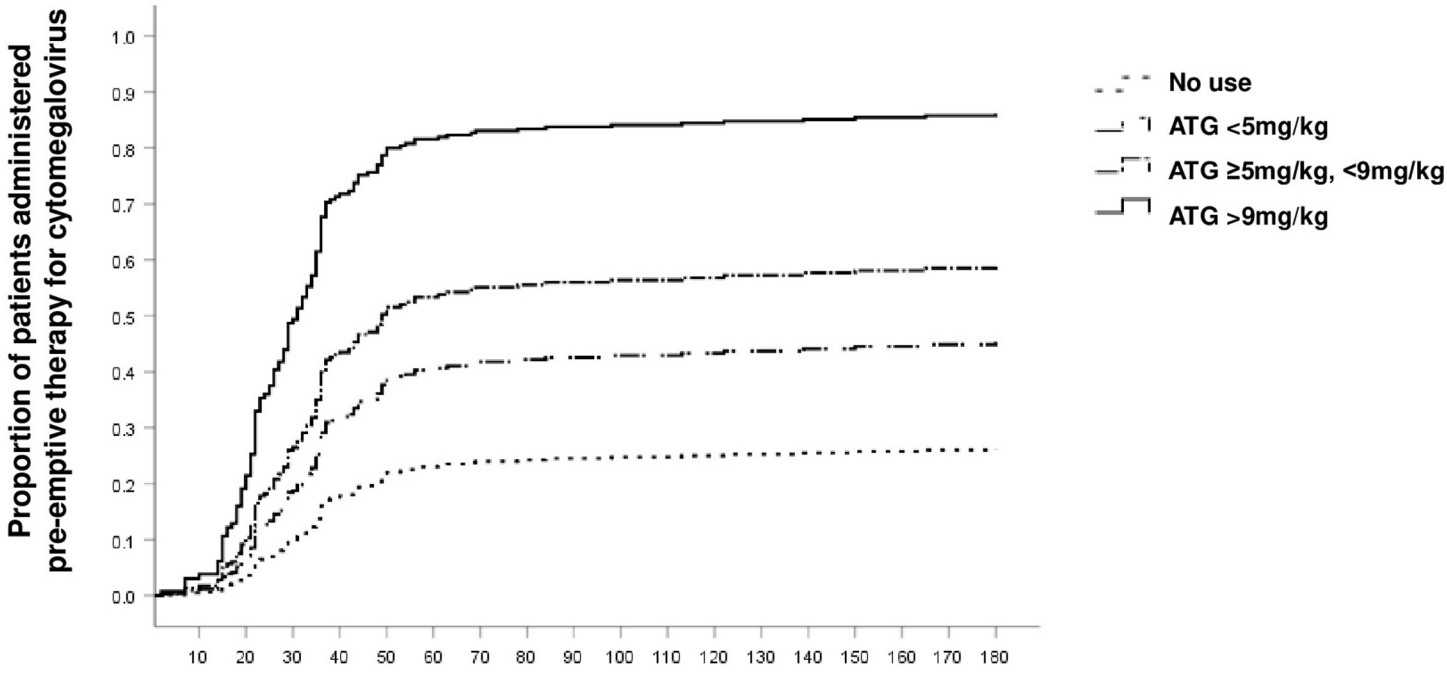

**Fig 2. Time to the start of pre-emptive therapy for CMV reactivation during allo-HSCT according to the use and dose of ATG.** Abbreviations: CMV, cytomegalovirus; allo-HSCT, allogeneic hematopoietic stem cell transplantation; ATG, anti-thymocyte globulin.

patients undergoing allo-HSCT, it is reasonable to minimize it in consideration of hospital stay, medical costs, and quality of life. In this study, the risk of CMV reactivation was associated with the diagnosis of multiple myeloma, CMV seropositivity of recipients and donors before allo-HSCT, acute GVHD grade ≥2, and ATG use and dose. Well-known CMV reactivation risk factors during allo-HSCT include CMV-positive recipient serostatus, acute GVHD grade ≥2 and its duration, and unrelated or mismatched donors [27, 29, 36–38]. In addition, the use of ATG [39], a higher titer of CMV-IgG in recipients before allo-HSCT [40], and the titer of the human leukocyte antigen allele type [41] have also been associated with CMV reactivation. ATG use or dose may increase CMV reactivation in patients with aplastic anemia and renal transplantation [42–45]. More aggressive CMV prophylaxis, such as letermovir, should be considered in patients with these risk factors, which present before allo-HSCT or may occur during allo-HSCT.

This study had several limitations. First, this study retrospectively analyzed data from a small number of patients. We aimed to present the pattern and effect of CMV reactivation on clinical outcomes in CMV-endemic areas; however, the results of this study did not represent the entire situation in CMV-endemic areas. In addition, although all patients in this study underwent allo-HSCT, they had various diseases, including aplastic anemia, myelodysplastic syndrome, leukemia, multiple myeloma, and lymphoma. Before allo-HSCT, each patient was treated with different types and durations of chemotherapy or immunosuppressive therapies. Therefore, this study alone cannot confirm the above conclusions, and additional studies are required to accurately classify the risk of CMV reactivation in each patient group. Third, we used operational definitions when defining the factors of acute or chronic GVHD that could potentially influence the interpretation of clinical endpoints results. In clinical practice,

GVHD diagnosis is not solely based on objective indicators. Therefore, the operational definition of patients who received systemic steroid therapy at a dosage of 1 mg/kg of actual body weight or higher for a minimum of 7 days or more is not considered inappropriate. However, patients with mild GVHD might have been included, and the possibility of misdiagnosis of organ involvement as GVHD due to other causes, such as CMV infections, cannot be completely ruled out. Although we controlled for each factor through multivariate analysis, the results should be interpreted within the limitations of the retrospective study design. Nevertheless, it is meaningful that this study presented the real-world pattern and survival in patients undergoing CMV monitoring and pre-emptive therapy for CMV reactivation during HSCT in a CMV-endemic area and suggested that high-risk patients can benefit from more aggressive CMV prophylaxis. This study presented real-world data that could serve as a basis for broadening the indication of primary prophylaxis for CMV reactivation and its use as secondary prophylaxis.

## Conclusion

Although half the patients with allo-HSCT were administered pre-emptive therapy for CMV, CMV reactivation did not affect their survival, indicating the advantages of pre-emptive therapy, even in CMV-endemic areas. These results show that aggressive CMV primary prophylaxis is not necessarily applicable to all patients, even in CMV-endemic areas. However, in patients with multiple myeloma, in whom CMV seropositivity is observed in the recipient and donor before allo-HSCT and those given a higher dose of ATG during conditioning, it may be better to use aggressive CMV primary prophylaxis. In addition, patients who experience CMV reactivation or acute GVHD requiring systemic steroid therapy during allo-HSCT should be considered for aggressive secondary CMV prophylaxis. The cost-effectiveness of more aggressive CMV prophylaxis should be re-evaluated in patients at high risk for CMV reactivation.

## Supporting information

**S1 File. STROBE statement—checklist of items that should be included in reports of observational studies.**
(DOCX)

**S1 Dataset.**
(XLSX)

## Author Contributions

**Conceptualization:** Byung Soo Kim, Hwa Jung Sung.

**Data curation:** Ka-Won Kang, Min Ji Jeon, Eun Sang Yu, Dae Sik Kim, Byung-Hyun Lee, Se Ryeon Lee, Chul Won Choi, Yong Park, Byung Soo Kim.

**Formal analysis:** Ka-Won Kang.

**Funding acquisition:** Ka-Won Kang, Hwa Jung Sung.

**Investigation:** Ka-Won Kang.

**Methodology:** Ka-Won Kang.

**Resources:** Ka-Won Kang, Min Ji Jeon, Eun Sang Yu, Dae Sik Kim, Byung-Hyun Lee, Se Ryeon Lee, Chul Won Choi, Yong Park, Byung Soo Kim.

**Supervision:** Chul Won Choi, Yong Park, Byung Soo Kim, Hwa Jung Sung.

Writing – **original draft:** Ka-Won Kang.

Writing – **review & editing:** Ka-Won Kang, Hwa Jung Sung.

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
