## [Decision Letter · Decision Letter 0]

28 Jun 2023

PONE-D-23-13734Cytomegalovirus reactivation under pre-emptive therapy during allogeneic hematopoietic stem cell transplant: pattern, survival, and risk factors in the Republic of KoreaPLOS ONE

Dear Dr. Sung,

Thank you for submitting your manuscript to PLOS ONE. After careful consideration, we feel that it has merit but does not fully meet PLOS ONE’s publication criteria as it currently stands. Therefore, we invite you to submit a revised version of the manuscript that addresses the points raised during the review process.

We have received the opinions of the expert reviewer and we invite you to submit a revised version of the manuscript. Please consider and address each of the comments raised by the reviewers.  

We look forward to receiving your revised manuscript.

Kind regards,

Senthilnathan Palaniyandi, Ph.D

Academic Editor

PLOS ONE

Reviewers' comments:

Reviewer's Responses to Questions

**Comments to the Author**

1. Is the manuscript technically sound, and do the data support the conclusions?

Reviewer #1: Yes

Reviewer #2: Yes

2. Has the statistical analysis been performed appropriately and rigorously? 

Reviewer #1: Yes

Reviewer #2: Yes

3. Have the authors made all data underlying the findings in their manuscript fully available?

Reviewer #1: Yes

Reviewer #2: Yes

4. Is the manuscript presented in an intelligible fashion and written in standard English?

Reviewer #1: Yes

Reviewer #2: Yes

5. Review Comments to the Author

Reviewer #1: In their manuscript Dr Kang and colleagues retrospectively analyzed the patterns and survival after CMV reactivation in patients undergoing pre-emptive therapy during allo-HSCT in CMV-endemic areas. They also assessed patients at high risk for CMV reactivation, and suggested that those patients might benefit from aggressive CMV prophylaxis. Their work has some significance, but further analysis of the data is needed, and the structure and language expression of the article need to be further modified, and some errors need to be corrected.

Please see attachment for detail.

Reviewer #2: Main flaw of this study is that it includes patients in pre-letermovir era and the results do not translate into current clinical practice. The authors have mentioned about letermovir in introduction but then do not continue this thread. Having real-life data on LMV we know that the problem of CMV reactivation is now vestigial and the number of CMV reactivation decreased by 90-95%. Bearing that in mind, the paper has little novelty.

Abstract:

Introduction: should be transplantation not transplant

Methods: post-HSCT not during HSCT

Conclusions duplicate the results and should be corrected.

Introduction:

1. Recipients should be added that it concerns stem cell recipients

2. CMV seropositive recipients have higher risk of CMV seronegative recipients only if blood products are CMV seronegative

3. Please clarify the term “aggressive” prophylaxis used in the context of letermovir prophylaxis (as this term does not appear in the literature) referring to a reliable source or replace with “primary prophylaxis”.

Methods:

1. Its not clear from this section whether the authors use letermovir in CMV primary prophylaxis; in introduction the authors mentioned that LMV should be used in almost all Korean stem cell recipients

2. CMV monitoring weekly or twice weekly, but how long? To day +100 or longer?

3. Ganciclovir was used untill CMV was undetectable; does it mean that it could be a continouos therapy?; what about an oral agent (valganciclovir) which is commonly used in clinical practice?

4. ATG horse or rabbit? For whom? For matched related recipients too?

5. Information provided in the last sentence of the first paragraph (“We accessed the data from January 2022 to December 2022 for the purpose of data collection and research”) is irrelevant.

Results:

1. Did the authors observe any CMV diseases in CMV reactivated patients? What includes the term „infections” as a cause of death

2. In patients with intestinal acute GVHD did the authors check for CMV colitis?

3. Max duration of first anti-CMV treatment was 73 days; does it mean that some patients were receiving ganciclovir for more than 2 months?

4. Table 2 only statistically significant values should be provided

5. I havent found information on how many patients developed acute GVHD, stage? Organ involvement? Treatment? Its crucial for CMV reactivation

6. Table 3 should be removed, no important information included

7. Table 4 only statistically significant data should be left

8. Table 5 to be removed

9. Did the authors still use the same anti-viral agent even when there was a repeated CMV reactivation? No one received foscavir or cidofovir?

6. PLOS authors have the option to publish the peer review history of their article (what does this mean?). If published, this will include your full peer review and any attached files.

Reviewer #1: **Yes: **Xiao-Jun Huang, Xu-Ying Pei

Reviewer #2: No

---

## [Author Response · Author response to Decision Letter 0]

24 Jul 2023

List of reviewers’ comments and point-by-point responses

In each response, page and line numbers refer to the revised manuscript.

In the revised manuscript, text changes are highlighted in yellow. 

Reviewer #1:

In their manuscript Dr Kang and colleagues retrospectively analyzed the patterns and survival after CMV reactivation in patients undergoing pre-emptive therapy during allo-HSCT in CMV-endemic areas. They also assessed patients at high risk for CMV reactivation, and suggested that those patients might benefit from aggressive CMV prophylaxis. Their work has some significance, but further analysis of the data is needed, and the structure and language expression of the article need to be further modified, and some errors need to be corrected.

1. Abstract: 

Introduction: should be transplantation not transplant

Methods: post-HSCT not during HSCT

Conclusions duplicate the results and should be corrected.

Response: Thank you for your careful review of our manuscript. As you suggested, we have modified this (page 2, lines 16, 19, and 30–33). In addition, following your recommendation, we have changed the title of the manuscript from “Cytomegalovirus reactivation under pre-emptive therapy during allogeneic hematopoietic stem cell transplant: pattern, survival, and risk factors in the Republic of Korea” to “Cytomegalovirus reactivation under pre-emptive therapy following allogeneic hematopoietic stem cell transplant: pattern, survival, and risk factors in the Republic of Korea.” We have also made necessary modifications to the related words in the manuscript. 

2. Introduction:

(1) Recipients should be added that it concerns stem cell recipients

Response: Thank you for your careful review of our manuscript. As you pointed out, we have modified this (page 3, lines 43 and 47).

(2) CMV seropositive recipients have higher risk of CMV seronegative recipients only if blood products are CMV seronegative

Response: Thank you for your careful review of our manuscript. As you suggested, we have modified this (page 3, lines 42–46).

(3) Please clarify the term “aggressive” prophylaxis used in the context of letermovir prophylaxis (as this term does not appear in the literature) referring to a reliable source or replace with “primary prophylaxis”.

Response: Thank you for your careful review of our manuscript. We have modified the sentence from “CMV primary prophylaxis is recommended in patients at high risk of CMV reactivation during allo-HSCT, with letermovir being considered aggressive primary prophylaxis in CMV-seropositive recipients” to “CMV primary prophylaxis is recommended in recipients of allo-HSCT at high risk of CMV reactivation following allo-HSCT, with letermovir being considered in CMV-seropositive recipients.” (page 3, lines 48–50).

3. Methods:

(1) It is not clear from this section whether the authors use letermovir in CMV primary prophylaxis; in introduction the authors mentioned that LMV should be used in almost all Korean stem cell recipients. 

Response: Thank you for the suggestion. In the Republic of Korea, letermovir has been covered by national insurance since August 2020. When we identified patients who met the indications for letermovir usage, most were recommended its use because the Republic of Korea is a CMV-endemic area. Therefore, we aimed to obtain objective information to identify the advantages and disadvantages of existing CMV pre-emptive therapies and why letermovir should be used in asymptomatic high-risk patients in this study. Consequently, patients who received letermovir as CMV primary prophylaxis were excluded from this study. The manuscript has been revised to ensure that this information is conveyed in the main text (page 3, line 58–59, page 4, lines 60, 62–63, and 73–76).

(2) CMV monitoring weekly or twice weekly, but how long? To day +100 or longer?

Response: Thank you for your careful review of our manuscript. We included only patients who had been monitored for CMV reactivation using the CMV antigen or CMV PCR for at least 6 months from the date of allo-HSCT. The manuscript has been revised to convey this information (page 4, lines 71–73, and page 6, lines 113–114).

(3) Ganciclovir was used untill CMV was undetectable; does it mean that it could be a continuous therapy?; what about an oral agent (valganciclovir) which is commonly used in clinical practice?

Response: Thank you for your careful review of our manuscript. Pre-emptive therapy for CMV was continuously administered with ganciclovir (5 mg/kg of actual body weight, through intravenous injection every 12 h) until a negative test result for the CMV antigen or a CMV PCR titer of <1,000 copies/mL was noted. The oral agent valganciclovir is available for use in the Republic of Korea but with insurance coverage issues in the study period. At our medical center, pre-emptive therapy for CMV was administered using the intravenous agent ganciclovir for all patients requiring pre-emptive therapy for CMV according to the treatment policy at the Korea University Medical Center. The manuscript has been revised to convey this information (page 6, lines 117–121).

(4) ATG horse or rabbit? For whom? For matched related recipients too?

Response: Thank you for the suggestion. In the Republic of Korea, only antithymocyte globulin rabbits are available. The manuscript has been revised to convey this information (page 5, lines 106–107, and page 6, lines 108–109).

We retrospectively analyzed data from three tertiary hospitals affiliated with the Korea University Medical Center (Anam, Guro, and Ansan) in the Korea University Transplant Registry. Rabbit anti-thymocyte globulin was used at the discretion of the participating physicians, depending on the type of donor and conditioning regimen. Reviewing the patients analyzed in this study revealed that even among matched related recipients, some received rabbit anti-thymocyte globulin (57/200 patients).

(5) Information provided in the last sentence of the first paragraph (“We accessed the data from January 2022 to December 2022 for the purpose of data collection and research”) is irrelevant.

Response: Thank you for your careful review of our manuscript. This content was requested by the editorial office during the initial submission of the manuscript. We have removed this information from the manuscript and moved the relevant information to an appropriate section. 

4. Results:

(1) Did the authors observe any CMV diseases in CMV reactivated patients? What includes the term “infections” as a cause of death. 

Response: Thank you for the suggestion. Infections were calculated as the mean of all forms of infection (bacterial, fungal, viral, etc.). Patients who died of infections were administered broad-spectrum antibiotics and antifungal agents until the day of death, making it difficult to conclusively attribute their deaths to CMV disease in such circumstances. Therefore, we presented patients who died from all forms of infection and the number of patients who received pre-emptive therapy for CMV reactivation using ganciclovir up to 1 week before their death. As you pointed out, we have revised the manuscript accordingly (page 11, lines 194–198).

(2) In patients with intestinal acute GVHD did the authors check for CMV colitis?

Response: Thank you for the suggestion. In this study, the occurrence of acute GVHD grade ≥2 or chronic GVHD with more than moderate severity was distinguished using an operational definition: patients who received systemic steroid therapy at a dosage of 1 mg/kg of actual body weight or higher for a minimum of 7 days or more. (We describe the details of the contents corresponding to the following (5) questions below). This was done to address missing values resulting from patient compliance issues (including inaccurate measurement of diarrhea quantity), the inability to conduct appropriate tests owing to a deteriorating condition, or insufficient information in the medical chart. However, in clinical practice, GVHD is not determined solely based on objective indicators. Therefore, although classifying patients based on systemic steroid use has limitations, it is considered appropriate.

 As you pointed out, distinguishing whether this is because of acute GVHD or CMV enteritis in patients with gastrointestinal problems is necessary to interpret the research results accurately. To verify this situation based on objective evidence, patients with acute GVHD who underwent upper or lower gastrointestinal endoscopy were examined, and patients with histologically confirmed CMV colitis were identified. Of the 292 patients, 99 (33.9%) experienced acute GVHD grade ≥2, of which 25 underwent upper gastrointestinal endoscopy and 30 underwent lower gastrointestinal endoscopy. In four patients, CMV enteritis was confirmed by pathological examination. However, all four patients simultaneously received treatment for both acute GVHD with systemic steroids and CMV reactivation during this period. Therefore, the exact causes of the gastrointestinal problems experienced by the patients could not be established. The overall impact on the results would be minimal because the number of relevant patients was small, and the main clinical endpoints underwent a multivariate analysis with adjustments for these factors. However, the potential limitations of this approach have been mentioned in the Discussion section (page 18, lines 290–299). 

(3) Max duration of first anti-CMV treatment was 73 days; does it mean that some patients were receiving ganciclovir for more than 2 months?

Response: Thank you for your careful review of our manuscript. The original dataset confirmed that one of the 139 patients who received the first pre-emptive therapy for CMV received treatment for 73 days. This is considered an extraordinary case. However, since the patient's data had no problems, we have presented the case without deleting it.

(4) Table 2 only statistically significant values should be provided.

Response: Thank you for the suggestion. We have made the necessary modifications to Table 2 according to your recommendations.

(5) I haven’t found information on how many patients developed acute GVHD, stage? Organ involvement? Treatment? It is crucial for CMV reactivation. 

Response: Thank you for the suggestion. This study was based on a retrospective chart review. Initially, we attempted to distinguish between patients who experienced acute GVHD grade ≥2 or chronic GVHD with more than moderate severity using data from medical charts according to existing definitions. However, this was difficult because of missing data. Therefore, we decided to switch the approach to distinguish patients with acute GVHD grade ≥2 or chronic GVHD of more than moderate severity. We have been treating patients with acute GVHD grade ≥2 or chronic GVHD with more than moderate severity using systemic steroid therapy with a dosage of at least 1 mg/kg of actual body weight or higher for more than a minimum of 7 days, according to the general guidelines of GVHD and treatment policy at the Korea University Medical Center. Therefore, based on the treatment policy of the Korea University Medical Center, we classified patients with acute GVHD grade ≥2 or chronic GVHD with more than moderate severity as those who received systemic steroid therapy at a dosage of at least 1 mg/kg of actual body weight or higher for more than 7 days. 

- To clearly express this aspect, we have revised the relevant content in the Methods section (page 7, lines 133–143).

- The number of patients with acute GVHD grade ≥2 or chronic GVHD of more than moderate severity is shown in Table 1. Additionally, relevant information has been included in the Results section of the manuscript (page 8, lines 174–176).

- In the Discussion section, we have mentioned the limitations arising from distinguishing patients with acute GVHD grade ≥2 or chronic GVHD with more than moderate severity using an operational definition (page 18, lines 290–299).

(6) Table 3 should be removed, no important information included.

Response: Thank you for the suggestion. We have deleted Table 3 and adjusted the manuscript content accordingly.

(7) Table 4 only statistically significant data should be left. 

Response: Thank you for the suggestion. We have made necessary modifications to Table 4 according to your recommendations. As Table 3 has been deleted, the table number has been changed from #4 to #3.

(8) Table 5 to be removed.

Response: Thank you for the suggestion. We have deleted Table 5 and adjusted the manuscript content accordingly.

(9) Did the authors still use the same anti-viral agent even when there was a repeated CMV reactivation? No one received foscavir or cidofovir?

Response: Thank you for the suggestion. In the Republic of Korea, foscavirs face cost-related issues, as they are not covered by insurance for CMV reactivation. However, cidofovir is available as a rare medicine only through the Korea Orphan & Essential Drug Center, which makes its distribution challenging. In this study, of 139 patients who experienced CMV reactivation, three received foscarnet for uncontrolled CMV reactivation, with no users of cidofovir. The reason why the number of patients who used these medications was limited is uncertain between the effective control of CMV reactivation with ganciclovir alone and cost or distribution issues preventing their use. Therefore, we did not describe the relevant information in the manuscript, as it could exaggerate the effect of pre-emptive therapy for CMV reactivation.

---

## [Decision Letter · Decision Letter 1]

25 Aug 2023

Cytomegalovirus reactivation under pre-emptive therapy following allogeneic hematopoietic stem cell transplant: pattern, survival, and risk factors in the Republic of Korea

PONE-D-23-13734R1

Dear Dr. Sung,

We’re pleased to inform you that your manuscript has been judged scientifically suitable for publication and will be formally accepted for publication once it meets all outstanding technical requirements.

Kind regards,

Senthilnathan Palaniyandi, Ph.D

Academic Editor

PLOS ONE

Additional Editor Comments (optional):

Reviewers' comments:

Reviewer's Responses to Questions

**Comments to the Author**

1. If the authors have adequately addressed your comments raised in a previous round of review and you feel that this manuscript is now acceptable for publication, you may indicate that here to bypass the “Comments to the Author” section, enter your conflict of interest statement in the “Confidential to Editor” section, and submit your "Accept" recommendation.

Reviewer #1: All comments have been addressed

2. Is the manuscript technically sound, and do the data support the conclusions?

Reviewer #1: Yes

3. Has the statistical analysis been performed appropriately and rigorously? 

Reviewer #1: Yes

4. Have the authors made all data underlying the findings in their manuscript fully available?

Reviewer #1: Yes

5. Is the manuscript presented in an intelligible fashion and written in standard English?

Reviewer #1: Yes

6. Review Comments to the Author

Reviewer #1: (No Response)

7. PLOS authors have the option to publish the peer review history of their article (what does this mean?). If published, this will include your full peer review and any attached files.

Reviewer #1: No

---

## [Editor Report · Acceptance letter]

4 Sep 2023

PONE-D-23-13734R1 

Cytomegalovirus reactivation under pre-emptive therapy following allogeneic hematopoietic stem cell transplant: pattern, survival, and risk factors in the Republic of Korea 

Dear Dr. Sung:

I'm pleased to inform you that your manuscript has been deemed suitable for publication in PLOS ONE. Congratulations! Your manuscript is now with our production department. 

Kind regards, 

on behalf of

Dr. Senthilnathan Palaniyandi 

Academic Editor

PLOS ONE